EMBO
Molecular Medicine

# mtDNA heteroplasmy level and copy number indicate disease burden in m.3243A>G mitochondrial disease

John P Grady[1,†,‡], Sarah J Pickett[1,*,‡], Yi Shiau Ng[1], Charlotte L Alston[1,2], Emma L Blakely[1,2], Steven A Hardy[1,2], Catherine L Feeney[1], Alexandra A Bright[1], Andrew M Schaefer[1], Gráinne S Gorman[1], Richard JQ McNally[3], Robert W Taylor[1,2], Doug M Turnbull[1] & Robert McFarland[1,**]

## Abstract

Mitochondrial disease associated with the pathogenic m.3243A>G variant is a common, clinically heterogeneous, neurogenetic disorder. Using multiple linear regression and linear mixed modelling, we evaluated which commonly assayed tissue (blood $N = 231$, urine $N = 235$, skeletal muscle $N = 77$) represents the m.3243A>G mutation load and mitochondrial DNA (mtDNA) copy number most strongly associated with disease burden and progression. m.3243A>G levels are correlated in blood, muscle and urine ($R^2 = 0.61–0.73$). Blood heteroplasmy declines by ~2.3%/year; we have extended previously published methodology to adjust for age. In urine, males have higher mtDNA copy number and ~20% higher m.3243A>G mutation load; we present formulas to adjust for this. Blood is the most highly correlated mutation measure for disease burden and progression in m.3243A>G-harbouring individuals; increasing age and heteroplasmy contribute ($R^2 = 0.27$, $P < 0.001$). In muscle, heteroplasmy, age and mtDNA copy number explain a higher proportion of variability in disease burden ($R^2 = 0.40$, $P < 0.001$), although activity level and disease severity are likely to affect copy number. Whilst our data indicate that age-corrected blood m.3243A>G heteroplasmy is the most convenient and reliable measure for routine clinical assessment, additional factors such as mtDNA copy number may also influence disease severity.

**Keywords** m.3243A>G; MELAS; mitochondrial disease; mtDNA copy number; mtDNA heteroplasmy

**Subject Categories** Genetics, Gene Therapy & Genetic Disease

## Introduction

The pathogenic mitochondrial DNA (mtDNA) m.3243A>G variant in the *MT-TL1* gene (encoding mt-tRNA[Leu(UUR)]) is the most common heteroplasmic mtDNA disease genotype (Goto *et al*, 1990; Elliott *et al*, 2008; Gorman *et al*, 2015, 2016). Disease burden and progression vary greatly between individuals, and the associated clinical spectrum is broad (Nesbitt *et al*, 2013). The cause of this heterogeneity is poorly understood; consequently, disease burden and long-term prognosis are very difficult to predict (Chinnery *et al*, 1997; Kaufmann *et al*, 2011; Mancuso *et al*, 2014; Weiduschat *et al*, 2014; Fayssoil *et al*, 2017).

In individuals harbouring m.3243A>G, mutant and wild-type mtDNA molecules coexist within the same cell, a situation termed *heteroplasmy*. Tissue segregation patterns of m.3243A>G vary widely; post-mitotic tissues such as muscle have higher levels of mutant mtDNA, whilst levels in blood decrease significantly over time (Sue *et al*, 1998; Rahman *et al*, 2001; Pyle *et al*, 2007; Rajasimha *et al*, 2008; Mehrazin *et al*, 2009; de Laat *et al*, 2012). Although patients with very severe disease tend to have a high proportion of mutant mtDNA, the relationship between heteroplasmy level and clinical phenotype is not straightforward and such *intra*-individual variation between tissues adds to the uncertainty (Chinnery *et al*, 1997; Fayssoil *et al*, 2017).

We aimed to determine which putative prognostic factors are most associated with m.3243A>G-related mitochondrial disease burden and progression. To answer this question, we characterised the levels of heteroplasmy and mtDNA copy number in three commonly sampled tissues: blood, urine and skeletal muscle, in 242 m.3243A>G carriers (including 195 symptomatic patients) from the MRC Mitochondrial Disease Patient Cohort UK. Using multiple linear regression, we ascertained which m.3243A>G heteroplasmy measure and mtDNA copy number were the most associated with total disease burden, as determined by the Newcastle Mitochondrial

1   Wellcome Centre for Mitochondrial Research, Institute of Neuroscience, Newcastle University, Newcastle upon Tyne, UK
2   NHS Highly Specialised Mitochondrial Diagnostic Laboratory, Newcastle upon Tyne Hospitals NHS Foundation Trust, Newcastle upon Tyne, UK
3   Institute of Health and Society, Newcastle University, Newcastle upon Tyne, UK
    *Corresponding author. Tel: +44 191 2085397; E-mail: sarah.pickett@ncl.ac.uk
    **Corresponding author. Tel: +44 191 2820340; E-mail: robert.mcfarland@ncl.ac.uk
    †Present address: Kinghorn Centre for Clinical Genomics, Garvan Institute, Sydney, NSW, Australia
    ‡These authors contributed equally to this work

Disease Adult Scale (NMDAS) (Schaefer *et al*, 2006). We used linear mixed modelling to investigate disease progression.

# Results

### m.3243A>G heteroplasmy measures in all three tissues are correlated

In the first instance, we sought to characterise the three measurements of heteroplasmy; all are significantly correlated. The strongest relationship is between blood and urine heteroplasmy levels (Fig 1A; $R^2 = 0.73$, $P < 0.001$; interaction with age included for a linear fit, $P < 0.001$), followed by muscle and blood heteroplasmy levels (Fig 1B; $R^2 = 0.64$, $P < 0.001$, interaction with age included, $P = 0.029$). Finally, urine and muscle levels are correlated (Fig 1C; $R^2 = 0.61$, $P < 0.001$, sex term included, see below). Urine levels are significantly lower than muscle (slope = 0.75, 95% CI = 0.60–0.91, $P < 0.001$).

### Blood and urine m.3243A>G heteroplasmy levels are negatively correlated with age

The correlation between muscle and blood heteroplasmy (Fig 1A) shows that younger individuals tend to have higher blood heteroplasmy levels, including age in the model increased $R^2$ from 0.38 to 0.64 ($P < 0.001$), consistent with previous reports of a decline in blood levels with age (Rahman *et al*, 2001; Pyle *et al*, 2007; Rajasimha *et al*, 2008). We therefore examined the relationship of all tissue heteroplasmy measures with age; both blood and urine heteroplasmy levels show significant negative correlations, although this is greater for blood than urine. Muscle heteroplasmy is not correlated with age (Fig 1D–F).

### Blood m.3243A>G heteroplasmy levels decline with time

To evaluate reports of an exponential decline in blood m.3243A>G heteroplasmy level, we examined levels in all patients with multiple measurements (Fig 2). Blood levels have declined in the majority of patients observed for 4 years or more but this is not universal and some individuals appear to reach a terminal nonzero plateau of heteroplasmy; therefore, decline is not universally exponential. The rate of heteroplasmy change in blood is not associated with disease burden ($P = 0.977$), and individuals whose level increased or remained stationary had a similar disease burden to those whose levels decreased.

We regressed the rate of heteroplasmy change against initial mutation level, estimating a continuous decline of −0.0185/year (95% CI = −0.0421 to 0.0052), consistent with previous methodology (Rajasimha *et al*, 2008). However, the model is a poor fit to our data ($R^2 = 0.06$, $P = 0.120$); this is improved if age at first measurement is included ($R^2 = 0.42$, $P = 0.002$), with older individuals having a lower rate of decline.

### Blood m.3243A>G heteroplasmy levels must be adjusted for age

To understand the decline in blood m.3243A>G levels across the whole cohort, we modelled the decline in blood heteroplasmy using adjusted urine heteroplasmy levels as an estimate of initial mutation

level ($N = 204$, see below) and validated our model using muscle heteroplasmy levels. We propose the following formula for age correction of blood heteroplasmy levels which represents a compound decline of ~2.3% a year with an added age adjuster to account for a rapid reduction in mutation load in early life (Appendix Supplementary Method 2).

Age-adjusted blood level = (Blood heteroplasmy)$/0.977^{(age+12)}$

Mean age-adjusted blood heteroplasmy is significantly correlated with mean muscle heteroplasmy (Fig 3A). This is an improvement on unadjusted blood levels ($R^2 = 0.48$ compared to 0.38), and the two levels can be more easily compared (slope = 1.03, 95% CI = 0.78–1.28). Adjusted levels are also correlated with urine heteroplasmy ($R^2 = 0.60$, $P < 0.001$, $N = 224$) and sex-adjusted urine heteroplasmy ($R^2 = 0.64$, $P < 0.001$, $N = 224$). There is a relationship between age-adjusted blood heteroplasmy and age (Fig 3B), but this is weaker and only seen in females.

### Urine m.3243A>G heteroplasmy levels are dependent on sex

The correlation between urine and muscle heteroplasmy level shows a clear effect of sex; males have, on average, 19.2% higher urine heteroplasmy (Fig 1C; 95% CI = 12.6–25.8%). Including sex in the model increases $R^2$ from 0.43 to 0.61 ($P < 0.001$). Consistent with this, a similar effect size is seen in Fig 1F; (male levels are 18.4% higher, 95% CI = 12.7–24.1%, $P < 0.001$). No relationship with sex is seen for blood ($P = 0.288$) or muscle heteroplasmy levels ($P = 0.245$).

### Urine m.3243A>G heteroplasmy levels must be adjusted for sex

Given the ~20% difference in m.3243A>G heteroplasmy levels between males and females, we used the relationship between urine and muscle levels to derive a method to adjust urine levels for sex (Appendix Supplementary Method 1).
Summary of transformation:

Male adjusted urine level = logit$^{-1}$((logit (Urine heteroplasmy)/ 0.791)−0.625)
Female adjusted urine level = logit$^{-1}$((logit (Urine heteroplasmy)/ 0.791)+0.608)

The adjusted levels are significantly correlated with muscle levels (Fig 3C), an improvement on the correlation between uncorrected urine and muscle levels ($R^2 = 0.55$ compared to 0.43; sex not included models). There is still a relationship with age, of a similar magnitude to that seen in Fig 1F (Fig 3D).

For 22 individuals, m.3243A>G mutation load in muscle is divergent from urine levels by > 20%; if adjusted urine levels are used, this drops to 12, and if adjusted blood levels are also taken into account, there are only five.

### Urine m.3243A>G heteroplasmy measures have the highest variability

We were interested in the *intra*-individual variability of measurements. Urine heteroplasmy levels appear to have much greater variability than the other tissues; levels in a number of individuals

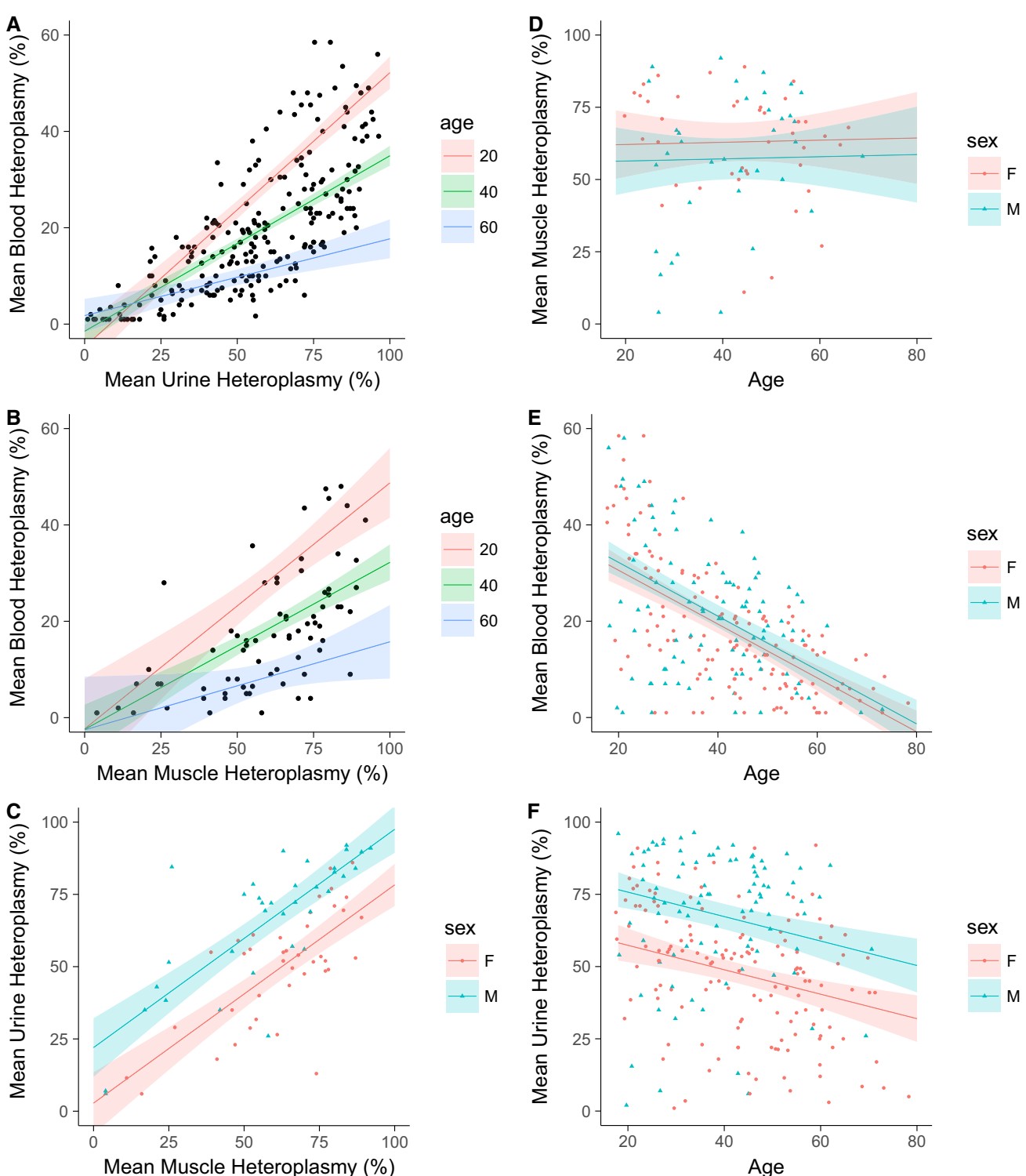

**Figure 1.  Correlations between heteroplasmy measurements and their relationship with age.**

A   Urine and blood heteroplasmy correlation ($N = 224$, $R^2 = 0.73$, $P < 0.001$; interaction with age included, $P < 0.001$)

B   Muscle and blood heteroplasmy correlation ($N = 74$, $R^2 = 0.64$, $P < 0.001$; interaction with age included, $P = 0.029$).

C   Muscle and urine heteroplasmy correlation, showing different intercepts for males and females ($N = 75$, $R^2 = 0.61$, $P < 0.001$).

D   The relationship between age and muscle heteroplasmy levels ($N = 77$, $R^2 = 0.0$, $P = 0.491$).

E   The relationship between age and blood heteroplasmy levels ($N = 231$, $R^2 = 0.32$, slope $= -0.56$, 95% CI $= -0.67$, $-0.45$, $P < 0.001$).

F   The relationship between age and urine heteroplasmy levels ($N = 235$, $R^2 = 0.22$, slope $= -0.42$, 95% CI $= -0.62$, $-0.22$, $P < 0.001$).

Data information: Points represent mean heteroplasmy level for each individual, and linear regression lines are shown with 95% confidence intervals.

differ by 20–33%, and one individual shows a 55% change. In contrast, the biggest change in blood is 15%.

To quantify the variability, we calculated the coefficient of variation (CV) for each individual with at least three repeated measurements, allowing us to compare the variability across all tissues (excluding muscle where $N = 1$). Age-adjusted blood shows the smallest variability ($N = 24$, mean CV = 0.091, 95% CI = 0.051–0.131), followed by unadjusted blood ($N = 24$, mean CV = 0.128, 95% CI = 0.084–0.171), although the difference is not significant ($P = 0.157$). The highest variability is seen in urine ($N = 39$, mean CV = 0.189, 95% CI = 0.148–0.230) and sex-adjusted urine ($N = 39$, mean CV = 0.213, 95% CI = 0.166, 0.259), which are significantly different from age-adjusted blood levels (both $P < 0.001$) and blood levels ($P_{urine} = 0.034$, $P_{sex-adjusted\ urine} = 0.015$).

### Total disease burden is most strongly associated with blood heteroplasmy level

To determine which heteroplasmy measurement is most strongly associated with total disease burden, we performed separate linear regression using each heteroplasmy measure. All are significantly associated, although the total variation explained by heteroplasmy, age and sex is low ($R^2$ range = 0.15–0.27; Table 1).

Age-adjusted blood (Fig 4A) and unadjusted blood heteroplasmy levels are more highly associated with total disease burden than adjusted and unadjusted urine levels (Table 1). The difference between using these two measures is small, failing to reach significance ($P = 0.173$; Appendix Table S1); however, as age was also included in the model, this is not surprising and provides further validation of our method of adjustment. Sex-adjusted urine level is the next most associated factor and better than uncorrected urine levels (Table 1), although this difference is not significant ($P = 0.166$; Appendix Table S1A). A composite measure of adjusted blood and urine m.3243A>G levels explained no more of the variance than using age-adjusted blood levels alone. Interestingly, blood heteroplasmy is significantly more associated with total disease burden than urine heteroplasmy ($P = 0.007$) and this holds when both corrected measures are used ($P = 0.036$; Appendix Table S1A).

In individuals with available muscle heteroplasmy data (Table 1), using blood or urine heteroplasmy levels are just as good as muscle (Appendix Table S1A).

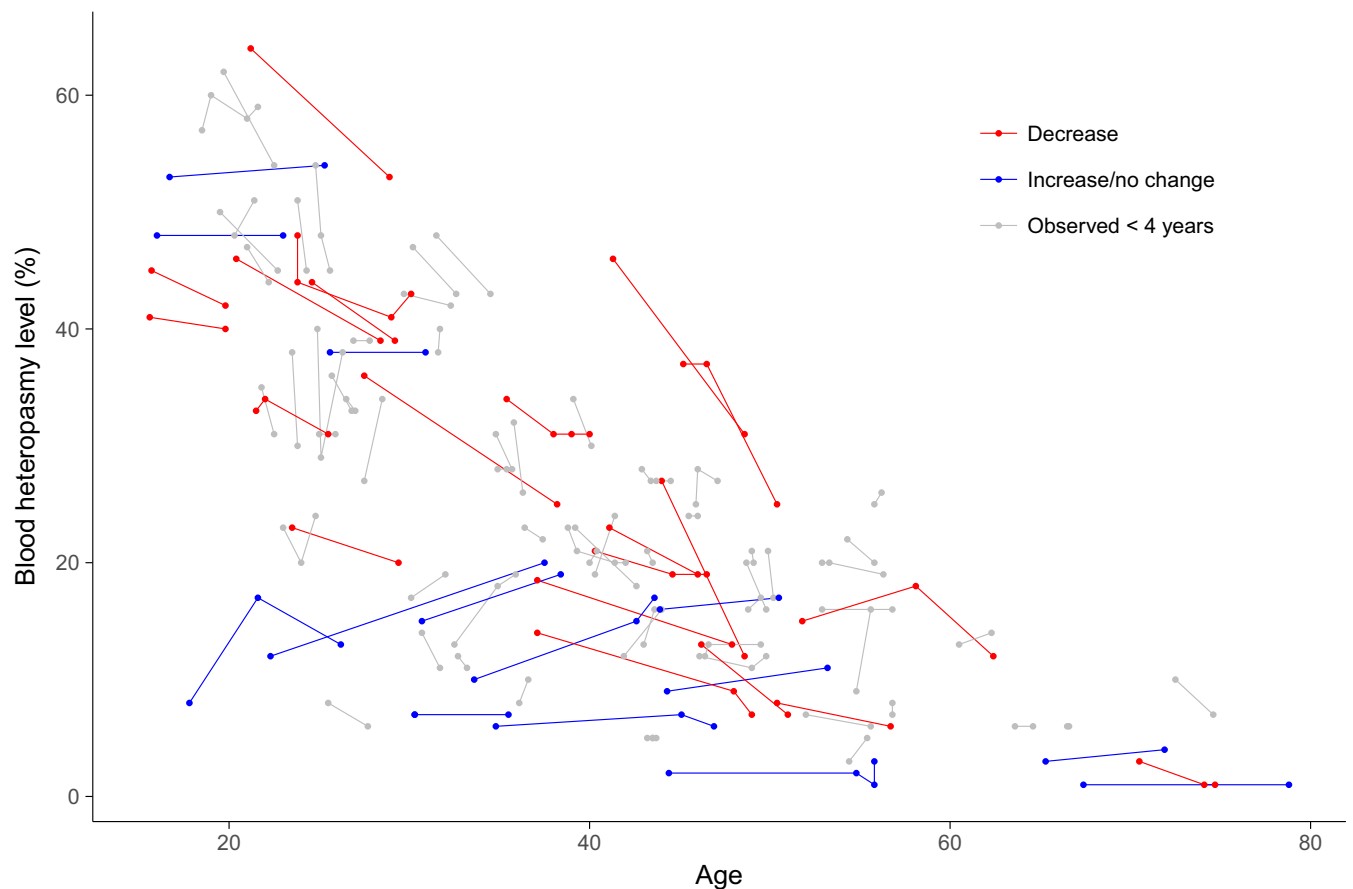

**Figure 2. Decline of blood m.3243A>G heteroplasmy level.**
Longitudinal blood heteroplasmy levels in individuals with multiple measurements ($N = 96$). Each point represents one heteroplasmy measurement; points joined by a line representing individual patients. Patients observed over 4 years or more are highlighted in blue for those showing a decline ($N = 21$) or red for those who show no change or an increase ($N = 14$).

   

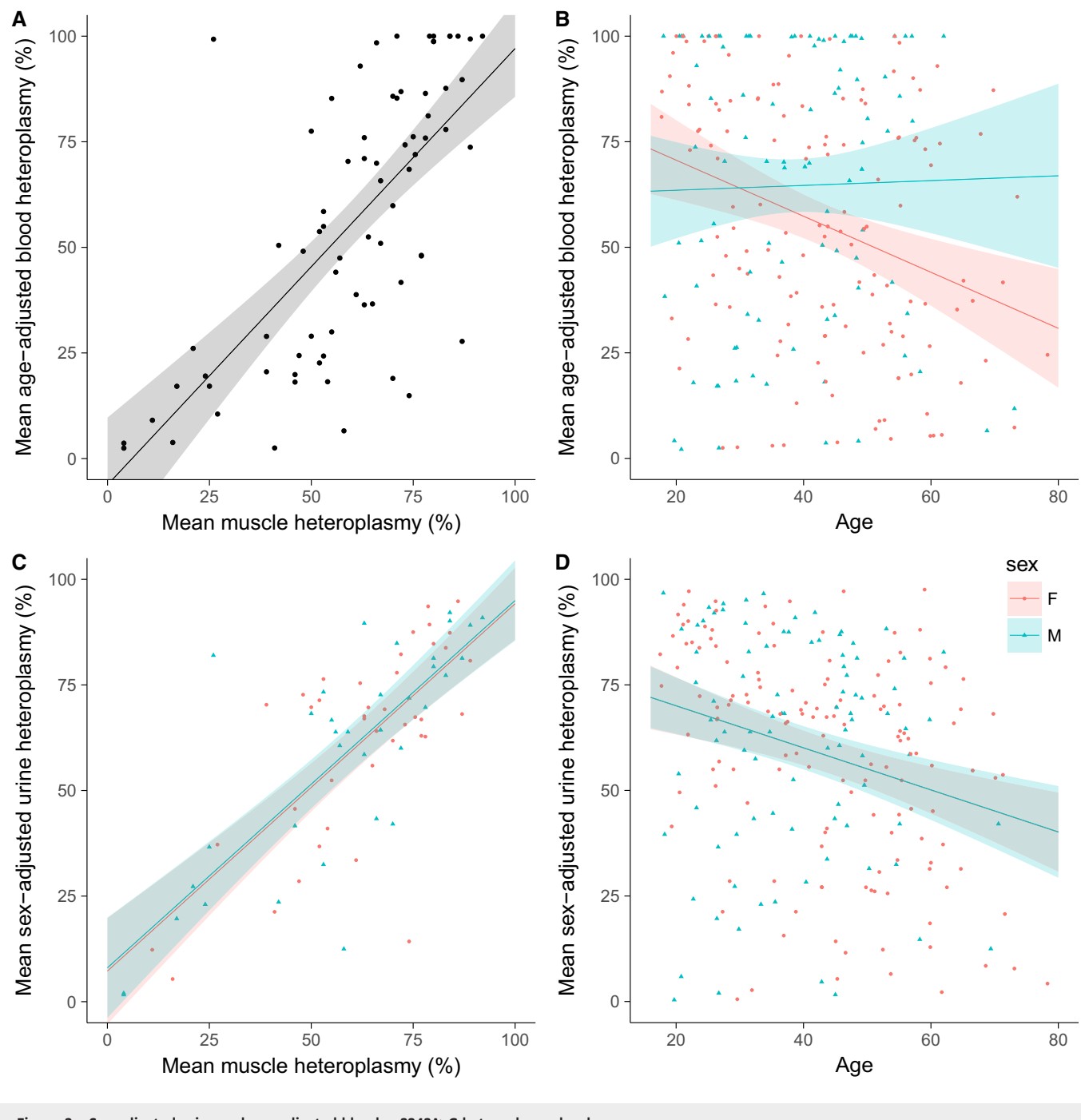

**Figure 3. Sex-adjusted urine and age-adjusted blood m.3243A>G heteroplasmy levels.**

A   Regression of age-adjusted blood heteroplasmy against muscle heteroplasmy ($N = 74$, $R^2 = 0.48$, $P < 0.001$).
B   Regression of age-adjusted blood heteroplasmy against age ($N = 231$, $R^2 = 0.06$, $P < 0.001$) showing an interaction with sex ($P = 0.022$).
C   Regression of sex-adjusted urine heteroplasmy against muscle heteroplasmy ($N = 75$, $R^2 = 0.55$, $P < 0.001$).
D   Regression of sex-adjusted urine heteroplasmy against age ($N = 235$, $R^2 = 0.07$, slope $= -0.50$, 95% CI $= -0.73$, $-0.27$, $P < 0.001$).

Data information: Linear regression lines and 95% confidence intervals are shown.

**Disease progression rate is most strongly associated with age-adjusted blood heteroplasmy level**

We then asked whether heteroplasmy level is associated with disease progression; we used a linear mixed model to examine the

role of heteroplasmy in the rate of NMDAS development. Disease progression is most strongly associated with age-adjusted blood levels, significantly more than urine ($P = 0.010$; Appendix Table S1B) and sex-adjusted urine levels ($P = 0.037$). Although the combination of adjusted blood and adjusted urine levels was

**Table 1.  Comparisons of different m.3243A>G heteroplasmy measures in association with total disease burden.**

| Heteroplasmy measure | $R^2$ | *P*-value |
|---|---|---|
| Larger cohort (N = 210) | | |
| Blood | 0.2744 | < 0.001 |
| Age-adjusted blood | 0.2539 | < 0.001 |
| Urine | 0.1809 | < 0.001 |
| Sex-adjusted urine | 0.1989 | < 0.001 |
| Mean adjusted blood and urine | 0.2461 | < 0.001 |
| Smaller cohort (N = 69) | | |
| Blood | 0.1547 | 0.001 |
| Age-adjusted blood | 0.1809 | < 0.001 |
| Urine | 0.1963 | < 0.001 |
| Sex-adjusted urine | 0.2005 | < 0.001 |
| Skeletal muscle | 0.2024 | < 0.001 |
| Mean adjusted blood and urine | 0.1955 | < 0.001 |

marginally more associated than adjusted blood levels alone, this difference failed to reach significance ($P = 0.624$). Crucially, in the cohort with available muscle heteroplasmy data, muscle m.3243A>G levels were not more strongly associated with disease progression than age-adjusted blood ($P = 0.189$), sex-adjusted urine ($P = 0.297$), unadjusted urine levels ($P = 0.320$) or a combination of adjusted blood and urine levels ($P = 0.696$).

Increasing age and the interaction between age and increasing age-adjusted blood heteroplasmy level are significantly positively associated with disease progression ($P < 0.001$); however, inter-individual variation is extremely high; the standard deviation of the variance due to individual is 2.35 (on NMDAS$^{0.5}$ scale) compared to 0.04 for age. Although individuals with high heteroplasmy levels tend to have a higher disease burden and rate of progression and vice versa, there is a large spread in the data as well as considerable overlap (Fig 4B).

**Total mtDNA copy number is affected by m.3243A>G heteroplasmy level (in blood) and sex (in urine)**

As only 27% of the total disease burden can be explained by m.3243A>G heteroplasmy level and age, we reasoned that the absolute quantity of wild-type mtDNA could affect clinical phenotype. We assessed mtDNA copy number in all three available tissues and determined the relationship between mtDNA copy number, sex, m.3243A>G heteroplasmy level and age. Standardised mtDNA copy number was highest in muscle (median = 3523, IQR = 1708, range = 678–10,610, $N = 66$) and lowest in blood (median = 144, IQR = 81, range = 24–323, $N = 197$), whilst urine mtDNA copy number showed the greatest range (median = 1181, IQR = 2870, range = 77–24,730, $N = 165$).

Blood mtDNA copy number shows a weak positive correlation with unadjusted ($R^2 = 0.03$, $P = 0.007$) and age-adjusted ($R^2 = 0.02$, $P = 0.016$) m.3243A>G heteroplasmy level ($N = 197$). We observed no association with sex ($P = 0.088$) or age ($P = 0.163$). Urine mtDNA copy number is significantly higher in males ($R^2 = 0.12$, $P < 0.001$, $N = 176$; median$_{male}$ = 3,733, IQR$_{male}$ = 5,630,

median$_{female}$ = 881, IQR$_{female}$ = 1,468), which is unsurprising given the differences seen in m.3243A>G heteroplasmy level and is likely to be due to differences in cellular composition. No association with age ($P = 0.628$), unadjusted ($P = 0.113$) or sex-adjusted ($P = 0.163$) urine heteroplasmy was observed. Muscle mtDNA copy number shows a slight downward trend with age, although this does not reach significance ($R^2 = 0.04$, $P = 0.060$, $N = 67$). No association with sex ($P = 0.450$) or muscle m.3243A>G heteroplasmy ($P = 0.725$) was detected. We saw no correlation between copy number in the three tissues studied (blood*-urine** $P = 0.372$, $N = 148$; blood*-muscle $P = 0.641$, $N = 59$; urine**-muscle $P = 0.493$, $N = 52$; *age-adjusted blood m.3243A>G heteroplasmy or **sex included in model).

**Low total mtDNA copy number in muscle is an indicator of disease burden**

To determine the effect of mtDNA copy number with disease burden, we performed separate linear regression models for each tissue, including age, m.3243A>G heteroplasmy level (assessed from the same sample as copy number) and sex (only for urine) as covariates. We found that higher skeletal muscle mtDNA copy number is significantly associated with reduced total disease burden in both single and multiple linear regression ($P_{mtDNA\ copy\ number} < 0.001$; $N = 66$; Fig 5). We estimate the effect of an increase in mtDNA copy number of 100 is a decrease in NMDAS$^{0.5}$ by 0.055 (95% CI = 0.079–0.031), which is similar to the effect of decreasing muscle m.3243A>G heteroplasmy by 1% (0.027, 95% CI = 0.008–0.046, $P = 0.006$). Indeed, mtDNA copy number, muscle heteroplasmy and age account for 40% of the variance in disease burden ($P < 0.001$); only 21% is explained when mtDNA copy number is not included. No significant associations were found for mtDNA copy number in either blood ($P_{blood} = 0.551$, $P_{age-adjusted\ blood} = 0.586$; $N = 197$) or urine ($P_{urine} = 0.144$, $P_{sex-adjusted\ urine} = 0.491$; $N = 165$). The interaction between age and muscle mtDNA copy number is associated with disease progression ($P = 0.0177$, $N = 63$), but no association was seen for blood ($P = 0.2491$, $N = 197$) or urine ($P = 0.1726$, $N = 159$).

## Discussion

We have characterised m.3243A>G heteroplasmy levels and mtDNA copy number in blood, urinary sediment and skeletal muscle, describing previously unreported sexual divergences in urine heteroplasmy and mtDNA copy number and confirming a longitudinal and continuous decline in blood heteroplasmy level in the majority of individuals. We have developed formulas to adjust urine m.3243A>G heteroplasmy for sex and have extended previously published methodology to adjust blood heteroplasmy levels for age, accounting for a higher initial rate of decline.

Our data show that age, muscle heteroplasmy level and muscle mtDNA copy number explain 40% of the variance in disease burden in m.3243A>G-related disease. This is approaching the 49% variance in disease progression explained in single, large-scale mtDNA deletion disease (Grady *et al*, 2014a). In contrast to previous reports, we do not see significant associations with mtDNA copy number in urine or blood and disease burden; however, our

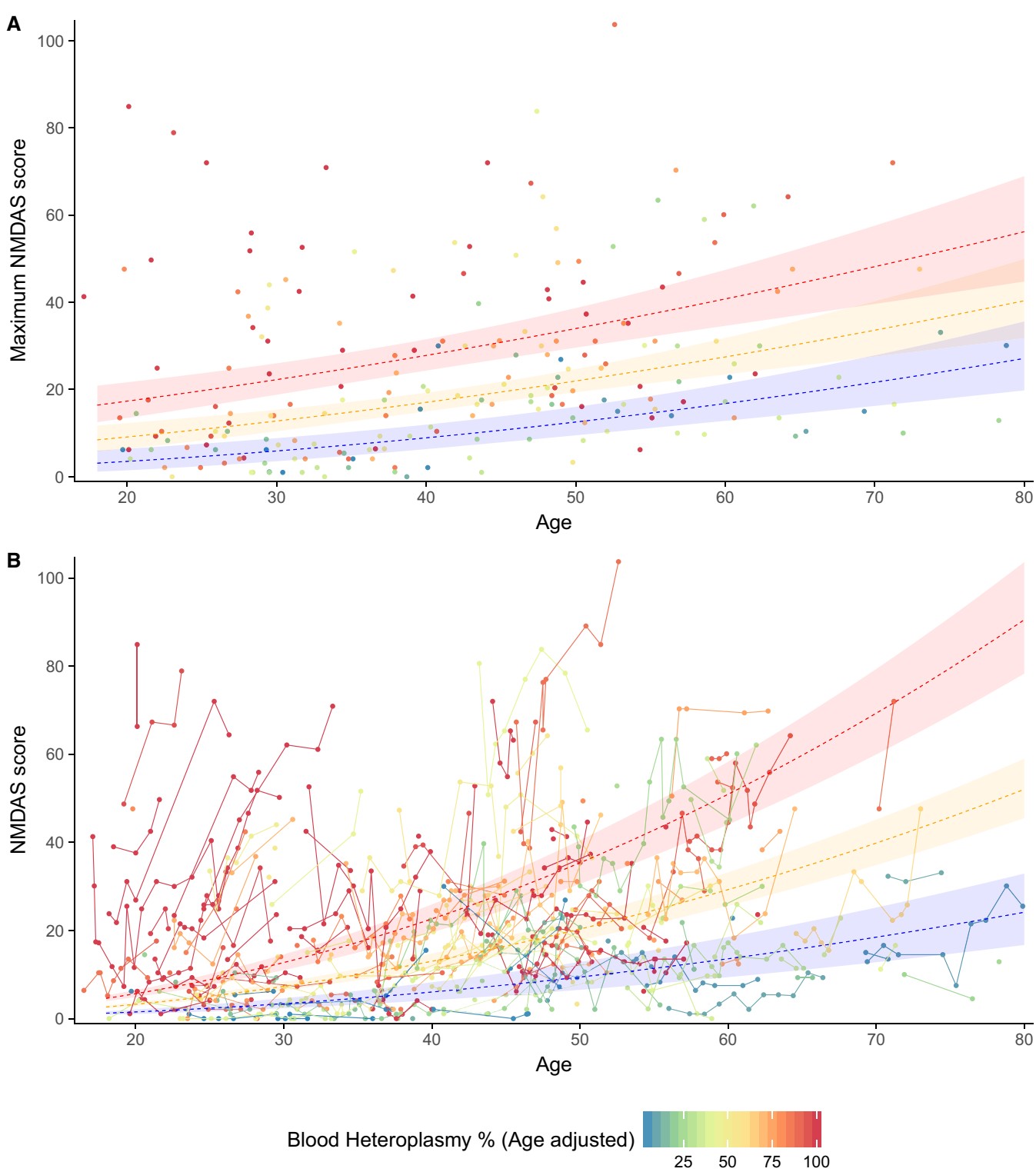

**Figure 4. Association of total disease burden and disease progression with age-adjusted blood heteroplasmy.**

Points are coloured according to mean age-adjusted blood heteroplasmy level. Dashed lines and shading represent scores and 95% confidence intervals predicted using a linear model (with the square root of NMDAS score as the dependent variable and age and age-adjusted blood heteroplasmy as independent variables) for individuals with 10% (blue), 50% (orange) and 90% (red) heteroplasmy levels.

A  Association of total disease burden with age-adjusted blood heteroplasmy. Points represent mean NMDAS score per individual.

B  Association of and disease progression with age-adjusted blood heteroplasmy. Each point represents one NMDAS assessment with points from each individual connected by a solid line (N = 210).

multi-variate models also include age and heteroplasmy, which may account for this (Liu *et al*, 2006, 2013). This result suggests that skeletal muscle is the most useful tissue to access and test in order to understand m.3243A>G-related disease.

Where copy number assessment is not possible, we show that muscle m.3243A>G heteroplasmy is not more highly associated with disease burden or progression than blood or urine. Both adjusted and unadjusted blood heteroplasmy levels are significantly more associated than adjusted and unadjusted urine levels. This contradicts previous studies (Mehrazin *et al*, 2009; Whittaker *et al*, 2009; de Laat *et al*, 2012) and may well reflect our larger sample size, inclusion of age as a risk factor, and the significantly higher within-individual variability that we observe in urine heteroplasmy levels.

Together with age, blood and age-adjusted blood heteroplasmy account for 25–27% of the variance in disease burden, consistent with previous estimates (Whittaker *et al*, 2009; de Laat *et al*, 2012). Individuals with high levels tend to have higher disease burden and progression, but, even using muscle mtDNA heteroplasmy and copy number, we observe meaningful *inter*-individual variation, highlighting the difficulties in providing accurate prognoses to patients

with m.3243A>G-related disease and implying that unidentified factors, such as nuclear genetic variation, environmental influences and epigenetics, affect clinical outcome (Picard & Hirano, 2016; Pickett *et al*, 2018). Efforts to identify these, as well as the mechanisms affecting mtDNA copy number and additional biomarkers for disease severity, will be crucial to understanding m.3243A>G-related mitochondrial disease.

The clear association with muscle mtDNA copy number and disease burden may simply reflect that more severely affected individuals are likely to have lower activity levels, which may reduce muscle mtDNA copy number (Apabhai *et al*, 2011; Cao *et al*, 2012). Alternatively, some individuals may have a compensatory mechanism of mitochondrial biogenesis triggered in the presence of mutant mtDNA. MtDNA copy number has been shown to influence the clinical outcome of patients carrying Leber hereditary optic neuropathy mtDNA mutations; asymptomatic carriers have a higher mtDNA copy number than both patients and healthy controls, with fibroblasts from carriers also displaying the highest capacity for activating mitochondrial biogenesis (Giordano *et al*, 2014). Studies in m.3243A>G cybrids have shown a positive correlation between mtDNA copy number and $O_2$ consumption, indicating that mtDNA

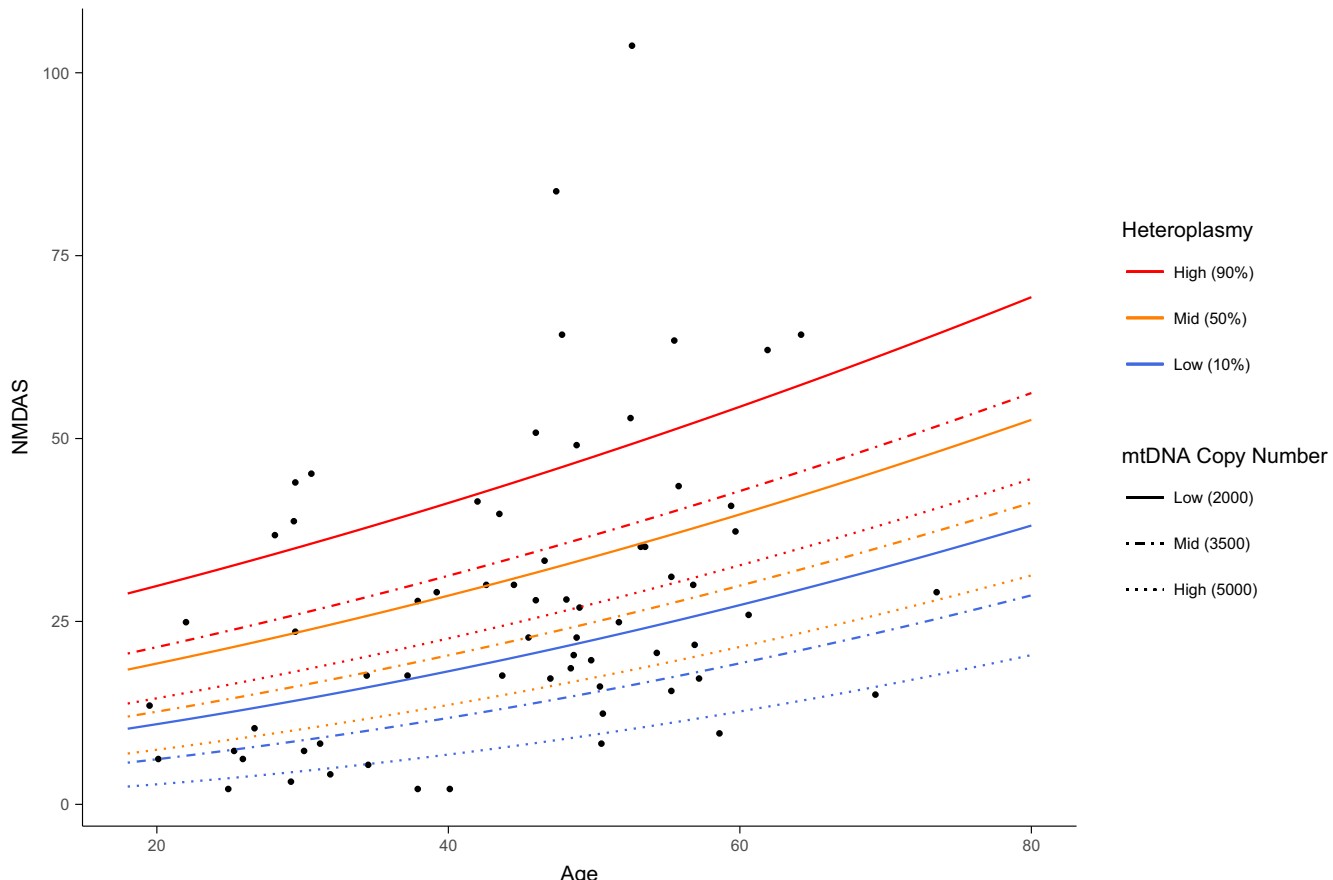

**Figure 5. Association of total disease burden with age, m.3243A>G heteroplasmy level and mtDNA copy number in skeletal muscle.**
Lines represent scores predicted using a linear model (with the square root of NMDAS score as the dependent variable and age, skeletal muscle heteroplasmy and mtDNA copy number as independent variables). Individuals with 10% (blue), 50% (orange) and 90% (red) heteroplasmy levels and low standardised mtDNA copy number (2,000 copies/nucleus; solid line), medium copy number (3,500 copies/nucleus; dashed line) and high copy number (5,000 copies/nucleus; dotted line) are represented.

copy number may be a limiting factor in the cellular mitochondrial respiratory phenotype within the context of this mutation (Bentlage & Attardi, 1996). Compensatory mechanisms could well be tissue-specific (we know that heteroplasmy plays a greater role for some tissue-specific phenotypes than for others) and determined by environmental or genetic factors, possibly accounting for some of the phenotypic variability seen in this disease (Pickett *et al*, 2018). It would therefore be fascinating to determine mtDNA copy number in other commonly affected tissues, although the availability of post-mortem nervous and cardiac tissue is likely to be restrictive. Our confirmation of the role of mtDNA copy number in m.3243A>G-related disease opens up the exciting possibility that components of the mitochondrial biogenesis pathway could be targeted by drugs in the treatment of this disease; indeed, change in skeletal muscle mtDNA copy number should be considered as an outcome measure for any clinical trial relating to m.3243A>G-related disease.

We characterised the decline in blood m.3243A>G heteroplasmy levels over time, estimating a compound decline of 2.3%/year, consistent with a previous estimate of 2% (Rajasimha *et al*, 2008). However, our data show more variability, with 14 of 35 individuals showing increasing or stationary blood heteroplasmy levels. There is also clear evidence of either an initial rapid decline, a slowing of decline with time, or both; evaluation of the relative contribution of each requires further study.

In some individuals, blood levels of the m.3243A>G mutation appear to drop to a nonzero and non-trivial proportion. Long-term follow-up is necessary to fully characterise this *inter*-individual variation in selection against the m.3243A>G mutation. These individuals did not have a significantly different disease burden, but we had limited power to detect an effect. Interestingly, four individuals (all > 48 years of age) had detectable mutation loads in urine but not blood, confirming that urine should also be checked for diagnosis if m.3243A>G is suspected.

Our proposed method for calculating age-adjusted blood m.3243A>G heteroplasmy produces levels better correlated with muscle heteroplasmy. A negative correlation with age is still present, although this is weaker and only in females. This is potentially attributable to ascertainment bias, reflecting our active tracing of m.3243A>G within pedigrees resulting in the recruitment of older, mildly affected, females who are more likely to have lower heteroplasmy levels than their more severely affected offspring.

The biological cause of the drop in m.3243A>G levels in blood remains unknown, but it is consistent with continuous selection against haematopoietic stem cells with high mutation levels, a theory supported by simulations although not by induced pluripotent stem cell studies, which show marked bimodal segregation towards homoplasmy (Rajasimha *et al*, 2008; Hamalainen *et al*, 2013; Kodaira *et al*, 2015). Along with the *inter*-individual variation in the rate of decline, this suggests that any putative selection process is not universal. Further studies to determine whether specific blood cell types are responsible for the reduction in level are warranted and may offer insight into the mechanism of selection.

Our sex-adjusted urine m.3243A>G heteroplasmy levels are better correlated with muscle levels than unadjusted levels, demonstrating that they are likely to be a better overall surrogate for whole organism heteroplasmy level. However, high *intra*-individual variability suggests that urinary sediment is not a reliable estimate of heteroplasmy level if taken in isolation.

The observed variability and sexual divergence of m.3243A>G heteroplasmy and mtDNA copy number in urinary sediment are likely to be due to differences in cellular content. Urine contains many different cell types, mostly epithelial in nature, and differences in cellular composition arise depending on sex, age and disease status (Benda *et al*, 2013). Other factors, such as time of day, volume, concentration or infection, could also alter cellular content and therefore influence measured heteroplasmy and copy number levels. If urine is to be pursued for diagnostic testing of m.34243A>G, variability in cellular content and the effect on heteroplasmy levels should be investigated. It is possible that this sexual divergence also exists in other tissues and may explain some of the sex-specific effects seen in mitochondrial disease (Giordano *et al*, 2014; Mancuso *et al*, 2014).

There are limitations to our study. We have not considered the baseline level of disease found in the general population in our model, although some features of m.3243A>G-related disease are common (e.g. migraine). However, from a clinical care perspective, overall disease burden is important, irrespective of aetiology. We also recognise that the NMDAS may be weighted towards particular phenotypes and so is not a perfect measure of disease burden; however, it is clinically validated and widely used, and thus the most appropriate tool currently available. Moreover, we have only considered three tissues and our sample size for muscle is considerably smaller than for urine and blood. Diagnostic facilities are limited to relatively easily sampled tissues and are not always representative of those commonly affected, which tend to be post-mitotic with high-energy demand (e.g. skeletal and cardiac muscle and central nervous tissue), but even in samples obtained from autopsy, the relationships between tissue-specific symptoms and heteroplasmy levels are not clear (Betts *et al*, 2006; Maeda *et al*, 2016; Picard & Hirano, 2016). Our study reflects the tissue availability in many diagnostic settings, making our results clinically applicable. Finally, our results represent mtDNA copy number and heteroplasmy in homogenate tissue; future work should determine these levels in individual muscle fibres to determine whether muscle fibre type has any influence. Taking account of this unknown and therefore uncontrolled variability may lead to a stronger correlation with disease burden.

This study demonstrates that although a substantial amount of variation in m.3243A>G-related disease burden remains to be explained, the best correlates are skeletal muscle mtDNA copy number, heteroplasmy and age, highlighting the importance of using all three measurements to understand this condition. As skeletal muscle mtDNA copy number is conceivably a marker for activity level, which is affected by disease severity, it may have limited utility in predicting likely disease burden. Blood m.3243A>G heteroplasmy levels are just as strongly associated with disease burden and progression as muscle heteroplasmy levels; therefore, we suggest that age-adjusted blood heteroplasmy is reliable and the most convenient heteroplasmy measure to use routinely within the clinical setting. Although urine may be a useful tissue for diagnosis of m.3243A>G, our results suggest that urine heteroplasmy levels should be adjusted for sex and are highly variable, and thus must be interpreted with caution. We have developed a web-based tool (http://www.newcastle-mitochondria.com/m-3243ag-heteroplasmy-tool/) to calculate adjusted blood and urine m.3243A>G mutation levels, supporting clinicians in their management of these patients.

Future research should focus on identifying pathways responsible for modulating mtDNA copy number in affected tissues, identifying additional factors responsible for the extensive clinical variability associated with this disease as well as collecting larger cohorts of muscle samples in order to study the effect of muscle mtDNA copy number on disease progression.

# Materials and Methods

### Study population

These studies were undertaken using data from 242 adult m.3243A>G carriers (147 females, Appendix Table S2) recruited into the MRC Mitochondrial Disease Patient Cohort UK between 2005 and 2017; all individuals were investigated by the NHS Highly Specialised Service for Rare Mitochondrial Disorders of Adults and Children in Newcastle upon Tyne. Patient follow-up was carried out (median interval = 1.1 years, IQR = 0.77) using the Newcastle Mitochondrial Disease Adult Scale (NMDAS), a validated scale to evaluate multi-system involvement and disease burden in adult mitochondrial disease (Schaefer *et al*, 2006). The cohort includes 195 symptomatic patients (NMDAS score ≥ 5), 30 asymptomatic carriers (NMDAS < 5) and 17 with incomplete NMDAS data.

The NMDAS assessment comprises 29 questions rated on a scale of 0–5; we calculated the sum of these (excluding the score for respiratory function, which is challenging to record accurately). NMDAS assessments with fewer than 25 of 29 responses were excluded. For those included, the total NMDAS score was divided by the number of questions answered and multiplied by 29 to derive a scaled NMDAS score.

Of the 242 individuals, 231 had at least one m.3243A>G heteroplasmy measurement in blood, 235 in urine and 77 in skeletal muscle. We excluded post-renal transplant urine heteroplasmy measurements from two patients after observing a large post-transplant reduction in mean urine heteroplasmy level (35 to 3%, N = 1).

For patients with multiple NMDAS assessments (N = 166), the median follow-up time was 3.75 years (IQR = 4.95, max = 11.9) with a median of three assessments per patient (IQR = 4, max = 16). The median age at last assessment was 43.7 years (IQR = 21.9, range = 19.7–79.9). The median maximum scaled NMDAS score in patients with all three heteroplasmy measurements available was 21.8 (IQR = 24.8, range = 1.0–103.7, N = 69) and in patients who only had urine and blood heteroplasmy measurements available was 16.6 (IQR = 23.8, range = 0–84.9, N = 141).

Informed consent was obtained from all subjects and that experiments conformed to the principles set out in the WMA Declaration of Helsinki and the Department of Health and Human Services Belmont Report.

### Heteroplasmy measurements

Total DNA was extracted from skeletal muscle (*tibialis anterior* or *vastus lateralis*), urinary sediment, and blood by standard procedures. Pyrosequencing on the Pyromark Q24 platform permitted quantification of m.3243G>A heteroplasmy levels, as previously described and validated (de Laat *et al*, 2016), with mutation-specific pyrosequencing primers according to GenBank Accession number NC_012920.1: 5′biotinylated forward: m.3143-3163; reverse: m.3331-3353; and reverse pyrosequencing primer: m.3244-3258 (IDT, Coralville, USA). The allele quantification application of Pyromark's proprietary Q24 software was used to calculate heteroplasmy levels (level of test sensitivity > 3% mutant mtDNA).

### mtDNA copy number assessment

mtDNA copy number was determined by real-time PCR using singleplex Taqman assays targeting mitochondrial MT-ND1 and nuclear B2M as previously described and validated (Grady *et al*, 2014b). Primers and probes were as follows: B2M (GenBank accession number: NG_012920) forward: n.8969-8990; B2M reverse: n.9064-9037; and 5′ 6-FAM labelled B2M probe n.9006-9032; MT-ND1 (GenBank accession number NC_012920.1) forward: m.3485-3504; MT-ND1 reverse: m.3553-3532; and 5′ VIC-labelled MT-ND1 probe m.3506-3529. Probes contained a non-fluorescent quencher and 3′ MGB moiety. DNA was serially diluted to ~10 ng/μl for B2M and ~0.1 ng/μl for ND1; 5 μl DNA was added to each 15 μl reaction. Each 96-well plate contained a standard curve derived from six tenfold serial dilutions of a plasmid containing both B2M and ND1 targets. $R^2$ values were > 0.999 and gradients fell between −3.3 and −3.6 for all standard curves. Each reaction was performed in triplicate; mean threshold cycle (Ct) and standard curves were used to determine the mtDNA copy number per nucleus for each sample. Within sample outliers and samples with Ct standard deviation Ct > 0.3 or mean Ct > 33 were excluded from the analysis. A control DNA sample included on each plate (coefficient of variation of copy number from 22 plates = 16.8%) was used to standardise mtDNA copy number to control for inter-plate variation using the following formula: standardised copy number = (copy number/on-plate control copy number) x mean control copy number.

### Statistical analysis

#### Linear regression
All statistical analyses were performed using the R statistical package (R Core Team, 2016). For initial investigations, the mean heteroplasmy level and age for each heteroplasmy subtype were utilised, yielding a single independent data point per individual. To quantify the relationships between different heteroplasmy measurements and heteroplasmy with age and time, linear regression was performed. Where appropriate, sex and age and interaction terms were tested for inclusion in models using the analysis of variance test (threshold $P < 0.05$). Model assumptions were checked by visual inspection of plots of residuals against fitted values, QQ and leverage (threshold; Cook's distance of < 1). We report adjusted $R^2$ values, $P$ values (3dp) and, where appropriate, slopes and 95% confidence intervals (CI). We report unadjusted $P$ values for reasons well documented in the literature and, particularly as we are testing *a priori* hypotheses with variables that are not all independent, this would be too conservative (Perneger, 1998).

#### Quantifying variability
In order to quantify the variability in heteroplasmy measurements, we calculated the standard deviation of measurements for each

**The paper explained**

**Problem**

The pathogenic m.3243A>G mitochondrial DNA variant, responsible for a highly heterogeneous neurogenic disorder, is thought to be present in up to one in 400 individuals. It is not only the most common mutation in our mitochondrial disease patient cohort but also the most problematic; giving accurate advice to patients regarding likely disease burden and progression remains incredibly challenging, nearly 30 years since this heteroplasmic pathogenic mtDNA variant was first described. A high mutation load is thought to be associated with more severe disease, but variation in heteroplasmy levels in commonly sampled tissues and a decline in levels in blood with age adds to the uncertainty.

**Results**

We have characterised m.3243A>G mutation load and mtDNA copy number in three commonly sampled tissues: blood, urinary sediment and skeletal muscle. m.3243A>G mutation load in urine has the highest variability and is about 20% lower in females compared to males; we present a method to adjust levels for this effect. We also extend previous methodology to adjust blood heteroplasmy levels for the age of the individual, accounting for non-uniform decline throughout a subject's lifetime. Blood m.3243A>G heteroplasmy levels are more strongly associated than urine levels; 27% of the variance in disease burden can be attributed to m.3243A>G heteroplasmy and age. Skeletal muscle mtDNA copy number, heteroplasmy and age together explain 40% although mtDNA copy number in muscle is likely to be affected by activity level and therefore disease severity.

**Impact**

Our evidence suggests that clinicians should use age-adjusted blood m.3243A>G mutation load, along with patient age, as an indicator of likely disease burden. If m.3243A>G mutation load data for urine are available, these must be adjusted for the divergence between male and female levels and interpreted with caution due to high variability. If available, skeletal muscle samples could also be used to assay mtDNA copy number, providing valuable information about mitochondrial biogenesis in m.3243A>G-related disease. As the proportion of explained variance remains low, further work to identify additional factors associated with disease burden and progression in patients harbouring the pathogenic m.3243A>G variant is warranted.

individual with at least three repeated measurements, adjusted for small N to remove estimator bias (Holtzman, 1950). We derived coefficients of variation (CV) by dividing by the population heteroplasmy mean, expressing variation as a proportion of the mean, therefore, allowing for the different population means between tissue heteroplasmy levels (Brown, 1998). We compared the mean and standard deviation of the CVs across the heteroplasmy measurements using two-tailed Wilcoxon rank sum tests with continuity correction.

*Disease burden*

For all analyses using NMDAS score, the square root was taken to ensure normality of the residuals. For total disease burden, a single independent point for each subject was determined by taking the maximum scaled NMDAS score and the earliest age at which this was reached. Separate linear regression using each of the different m.3243A>G heteroplasmy measures was performed, including age and heteroplasmy as predictors of square root scaled NMDAS score. To compare models, the difference between $R^2$ values was

calculated as a measure of improvement in the model. Significance ($P$ values representing the probability of the second model being better than the first) and 95% CIs of the difference in $R^2$ values were determined by bootstrapping (1,000 replicates).

*Disease progression*

To quantify the association between m.3243A>G heteroplasmy, age and progression of disease burden, we used nlme (Pinheiro *et al*, 2016) to perform a linear mixed effects analysis of the relationship between heteroplasmy level, age and NMDAS scaled score. As a fixed effect, we entered age into the model. As random effects, we had by-subject random slopes for the effect of age and the interaction between age and heteroplasmy level (heteroplasmy on its own was dropped from the model as $P > 0.05$) and no intercept (making the assumption that individuals were asymptomatic at birth). Two-sided $P$ values were obtained from the t-distribution reported in nlme. To assess which m.3243A>G heteroplasmy measure was most highly associated with disease burden, we selected the model with the lowest Akaike information criterion (AIC). Bootstrapping with 1,000 replicates determined the mean difference between AIC values, 95% CIs and significance. Reported $P$ values represent the number of samples with an AIC difference less than zero (representing the first model being a better fit than the second model).

**Expanded View** for this article is available online.

## Acknowledgements

We thank the patients who participated in this study and the Mitochondrial Disease Patient Cohort administrative team. We are grateful to Dr. Mark Roberts (Salford Royal NHS Foundation Trust), Dr. Maria E. Farrugia and Dr. Richard K Petty (Queen Elizabeth University Hospital, Glasgow), Dr. David Dick, (Norfolk and Norwich University Hospital), Dr. Mellisa Maguire (The Leeds Teaching Hospitals NHS Trust), Dr. Colin Mumford (Western General Hospital, Edinburgh) and Dr. Cheryl Longman (NHS Greater Glasgow and Clyde) for patient referral. We would also like to thank the anonymous reviewers whose insightful comments have improved the manuscript, to Dr. Helen A. Tuppen for technical assistance with the mtDNA copy number assay and to Mr. Doug Jerry for cohort data entry. Ethical approval was granted by the Newcastle and North Tyneside Research Ethics Committee (13/NE/0326). This work was supported by a Wellcome Trust Fellowship (204709/Z/16/Z) to SJP, the Wellcome Centre for Mitochondrial Research (203105/Z/16/Z), the Medical Research Council (MRC) Centre for Translational Research in Neuromuscular Disease, the Mitochondrial Disease Patient Cohort (UK) (G0800674), the Lily Foundation, the UK NIHR Biomedical Research Centre for Ageing and Age-related Disease Award to the Newcastle upon Tyne Foundation Hospitals NHS Trust and the UK NHS Highly Specialised Service for Rare Mitochondrial Disorders of Adults and Children. CLA was supported by a National Institute for Health Research (NIHR) doctoral fellowship (NIHR-HCS-D12-03-04). YSN holds an NIHR clinical lectureship. The views expressed in the submitted article are the authors' own and not an official position of the institution or funder.

## Author contributions

Conception and design of the study: JPG, SJP, RMcF, RWT, DMT, RJQMcN. Acquisition and analysis of data: SJP, JPG, YSN, CLA, ELB, SAH, AMS, GSG, CLF, AAB, RMcF. Drafting the manuscript: SJP. Critical review of manuscript: JPG, RMcF, RWT, DMT, GSG, RJQMcN, YSN.

## Conflict of interest

The authors declare that they have no conflict of interest.

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
