## [Review Process File · EMBO Molecular Medicine]

mtDNA heteroplasmy level and copy number indicate disease burden in m.3243A>G mitochondrial disease

John P Grady, Sarah J Pickett, Yi Shiau Ng, Charlotte L Alston, Emma L Blakely, Steven A Hardy, Catherine L Feeney, Alexandra A Bright, Andrew M Schaefer, Gráinne S Gorman, Richard J Q McNally, Robert W Taylor, Doug M Turnbull, Robert McFarland

Review timeline:

Submission date:	14 July 2017
Editorial Decision:	08 August 2017
Revision received:	15 February 2018
Editorial Decision:	27 March 2018
Revision received:	29 March 2018
Accepted:	04 April 2018

Editor: Roberto Buccione/Céline Carret

Transaction Report:

1st Editorial Decision

08 August 2017

Thank you for the submission of your manuscript to EMBO Molecular Medicine. I apologise for the delay in reaching a decision. In fact, we experienced significant difficulties in securing expert and willing reviewers in part due to the overlapping holiday season.

Although I was hoping to obtain a third evaluation, I am now proceeding based on the two consistent evaluations obtained so far as further delays cannot be justified.

You will see that although both Reviewers are appreciative of the technical quality of your work and underline its potential interest, a few critical and partially overlapping concerns are raised.

Reviewer 1 questions the clinical relevance of the findings at this stage of development and suggests that data showing the correlations between specific phenotypes and heteroplasmy are required. Reviewer 1 also points to the lack of conceptual advance, especially considering the limited news value in the manuscript.

Reviewer 2 is perhaps less reserved in terms of the immediate clinical applications of your work and suggests that at the very least, assessment of mtDNA copy number as a surrogate marker of mt biogenesis, and re-assessment of the statistical models by adding this variable are needed to confer some much-needed molecular and conceptual content to the manuscript.

Finally, both reviewers would have clearly liked to see more insightful discussion on various points.

In conclusion, while publication of the paper cannot be considered at this stage, we are willing to consider a substantially revised manuscript, provided however, that the Reviewers' concerns are fully addressed with further experimentation where required, especially concerning reviewer 2's request on mtDNA copy number. This course of action was confirmed during our cross-commenting exercise, together with the request to improve the stratification of correlations with the clinical phenotypes.

Please note that it is EMBO Molecular Medicine policy to allow a single round of revision only and that, therefore, acceptance or rejection of the manuscript will depend on the completeness of your responses included in the next, final version of the manuscript.

As you know, EMBO Molecular Medicine has a "scooping protection" policy, whereby similar findings that are published by others during review or revision are not a criterion for rejection. However, I do ask you to get in touch with us after three months if you have not completed your revision, to update us on the status. Please also contact us as soon as possible if similar work is published elsewhere.

EMBO Molecular Medicine now requires a complete author checklist (<http://embomolmed.embopress.org/authorguide#editorial3>) to be submitted with all revised manuscripts. Provision of the author checklist is mandatory at revision stage; The checklist is designed to enhance and standardize reporting of key information in research papers and to support reanalysis and repetition of experiments by the community. The list covers key information for figure panels and captions and focuses on statistics, the reporting of reagents, animal models and human subject-derived data, as well as guidance to optimise data accessibility.

We now mandate that all corresponding authors list an ORCID digital identifier. You may acquire one through our web platform upon submission and the procedure takes <90 seconds to complete. We also encourage co-authors to supply an ORCID identifier, which will be linked to their name for unambiguous name identification.

Please carefully adhere to our guidelines for authors (<http://embomolmed.embopress.org/authorguide>) to accelerate manuscript processing in case of acceptance.

I look forward to receiving your revised manuscript in due time.

***** Reviewer's comments *****

Referee #1 (Comments on Novelty/Model System):

This is a technically solid report which correlates m.3243A>G mutation load in three body samples versus clinical severity and progression in a large cohort. The data largely confirm prior studies, but also provides more sophisticated statistical framework to correlate heteroplasmy with clinical phenotype. Unfortunately, the findings are not very robust and do not provide large conceptual advances that are typically required for publication in EMBO Mol Med. Major revisions with additional data showing more robust correlations between specific phenotypes with heteroplasmy would be necessary to warrant publication in this journal.

Referee #1 (Remarks):

The manuscript by Grady and colleagues provides an overview of 242 adult carriers of the m.3243A>G mitochondrial DNA (mtDNA) mutation followed at Newcastle-Upon-Tyne. Individuals were evaluated clinically with the Newcastle Mitochondrial Disease Adult Scale (NMDAS) and molecularly with measurements of mutation heteroplasmy in blood (231 individuals), urine (235), and skeletal muscle (77). The major findings were: 1) decline in blood heteroplasmy by 2.3% annually with greater reductions in young individuals; 2) higher mutation load in urine of men compared to women; and 3) relatively high correlation between disease burden/progression and age-adjusted blood heteroplasmy compared to mutation levels in urine and muscle. Nevertheless, the contribution of age and heteroplasmy to disease burden/progression was low (R-squared=0.25) indicating that at least one additional factor is responsible for clinical severity. Furthermore, the low inter-individual correlation between clinical severity and mutation load precludes predictions of future disease progression/severity based on measurements of heteroplasmy. The manuscript is written clearly, results appear reliable, and conclusions are supported by the data. The findings extend previous reports demonstrating decline of m.3243A>G

mutation load in blood, higher heteroplasmy in urine and muscle versus blood, and large variability in mutation burden relative to clinical severity. The authors have admirably better quantitated the decline in mutation load in blood and relative contribution of age and mutation burden in limited available clinical samples to disease progression.

Major comments

- 1) The title (which fails to mention m.3243A>G heteroplasmy) must be revised because it does not accurately reflect the contents of the paper.
- 2) A table should be added to provide the overall demographics of the study cohort (number of men vs. women, numbers of patients stratified by ages, ethnicities [if any variability], etc).
- 3) The authors should assess potential correlation of a composite age-adjusted blood and urine mutation heteroplasmy compared to disease burden/progression (i.e. age-adjusted mean of blood and urine heteroplasmy compared to NMDAS score).
- 4) Was m.3243A>G mutation load assessed in buccal swabs or saliva?
- 5) As noted by the authors, only overall disease burden was assessed in this study and a future study of heteroplasmy versus "specific phenotypic features would be valuable". Nevertheless, the authors should consider in the current analysis, muscle heteroplasmy with muscle weakness/functional impairment and urine mutation load with renal involvement.
- 6) In Figure 3B, it appears that the age-adjusted decline in blood heteroplasmy load is due to the effect in women and not in men.
- 7) The threshold $P < 0.05$ for analysis of variant tests is not appropriate due to the multiple tests performed. A correction for multiple test is warranted.

Minor Comments

- 1) Abstract: Change "Age-adjusted blood is the most highly correlated heteroplasmy measure" to "Age-adjusted blood heteroplasmy is the most highly correlated mutation measure"
- 2) Abstract: Change "...indicating that unidentified factors" to "indicating that at least one unidentified factor"
- 3) Results, page 4: Change "exponential" to "continuous".
- 4) Figure citations appear to be incorrect on page 5: "Figure 3C" should be "3B" while "Figure 3B" should be "3C".

Referee #2 (Comments on Novelty/Model System):

On the molecular ground this study lacks one variable that could have been assessed by these experienced group of researchers in mitochondrial medicine, i.e. the mtDNA copy number. Given the feasibility of this experiment, if performed and correlated to the currently existing data, it would increase substantially the novelty.

Referee #2 (Remarks):

The study by Grady and colleagues tackles an unresolved issue relevant both at the clinical and basic mitochondrial biology levels, i.e. the poor correlation of tissue heteroplasmy and disease burden with the common MELAS mutation at position 3243 tRNA^{Leu}. To get insights on this issue the authors systematically analyzed the heteroplasmic mutational load in DNA derived from three common sources in clinical settings: blood, urinary sediment and skeletal muscle biopsies. In a subset of cases they had multiple time-points for some of the tissues considered, specifically for blood and urinary sediment samples, which allowed for assessment of intra-individual consistency. Furthermore, all results are analyzed taking into consideration two other factors, sex and age, which have been previously shown to impinge on variability of heteroplasmy in a tissue-specific fashion, such as, for example, the age-dependent decline of mutant loads in blood heteroplasmy. A number of statistical manipulations, adjustments and modeling are applied to correlate the heteroplasmy data amongst them and with disease burden, as measured by the NMDS scale. The study reaches a few interesting results, which will be helpful in clinical management of patients. They can be summarized as:

1. the strongest correlation is between blood and urine heteroplasmy; urine levels are significantly lower than muscle
2. blood and urine levels are negatively correlated with age; the age-related decline of mutant loads

is not linear, has faster rate at young age and lower at older age

3. after age-adjustments blood levels correlate also with muscle; there is a clear effect of gender on urine levels of heteroplasmy, males having higher mutant load than females; furthermore, urine assessment of heteroplasmy is far more variable over repeated sampling than blood or muscle, thus one single assessment has poor reliability

4. total disease burden and progression are, a bit surprisingly, most highly correlated with blood heteroplasmy, either adjusted or non adjusted.

The latter finding at point 4 is the most relevant for clinical practice introducing the idea that to be informed on clinical prognosis blood DNA is enough, avoiding the invasive procedure of muscle biopsy.

Overall, I agree that this study clarifies a lot of points, some still controversial, and helps to understanding the clinical expression of the so-called MELAS mutation, thus representing a substantial improvement on previous smaller and non definitive studies.

I do have some critical comments though. I noticed that amongst the limitations that the authors mention, there is one that is something that they could/should have done in my opinion. On the molecular ground, and being the journal of interest EMBO Molecular Medicine, the authors should/could have assessed the mtDNA copy number as surrogate marker of mitochondrial biogenesis, to introduce a further element that may help to interpret the still poor and incomplete correlation of heteroplasmy with disease burden and progression rate. There is clear evidence that heteroplasmy and mtDNA copy number are tightly associated in determine the pathogenicity of the MELAS mutation. Giuseppe Attardi did show this in cybrids a long time ago (Bentlage HA, Attardi G. Hum Mol Genet. 1996 Feb;5(2):197-205), and recent studies on LHON strikingly highlighted how mtDNA copy number and compensatory activation of mitochondrial biogenesis may determine penetrance (Giordano et al., Brain. 2014 Feb;137(Pt 2):335-53). In the case of MELAS mutation, mtDNA copy number may well modulate clinical expression determining disease burden and rate of progression. Furthermore, the efficiency of compensatory activation of mitochondrial biogenesis that is reflected in mtDNA copy number is individually very variable, with individuals more efficient or less efficient in executing this response. Thus, mtDNA copy number is a molecular variable that seems crucial to fully understand the heteroplasmy issue.

I insist on this point because the authors have the DNA samples and the technical skills to run the experiment. There are issues on having good quality data for mtDNA copy number quantitative assessment based on the DNA extraction methods used, way of DNA conservation etc., all factors well-known to obtain reliable results.

In conclusion, apart the issue of mtDNA copy number, I believe the authors squeezed as much as possible from the kind of material and data they had available, with appropriate methods and statistical analysis. May be I would have liked to see a bit more reasoning on some of the results. For example, is there any hypothesis to explain the gender difference on urine heteroplasmy, considering that gender recently emerged as a relevant factor in a large survey of Italian patients carrying the same mutation (Mancuso et al., J Neurol. 2014 Mar;261(3):504-10), and is well known in LHON again, possibly correlated with the metabolic effect of estrogens on mitochondrial biogenesis (Giordano et al., Brain. 2011 Jan;134(Pt 1):220-34).

1st Revision - authors' response

15 February 2018

Responses to Reviewer 1

Summary:

Reviewer 1 questions the clinical relevance of the findings at this stage of development and suggests that data showing the correlations between specific phenotypes and heteroplasmy are required.

Reviewer 1 also points to the lack of conceptual advance, especially considering the limited news value in the manuscript.

Comment 1:

The title (which fails to mention m.3243A>G heteroplasmy) must be revised because it does not accurately reflect the contents of the paper.

Response 1:

Thank you for drawing this to our attention. We have amended the title to:

“mtDNA heteroplasmy level and copy number indicate disease burden in m.3243A>G mitochondrial disease.”

Comment 2:

A table should be added to provide the overall demographics of the study cohort (number of men vs. women, numbers of patients stratified by ages, ethnicities [if any variability], etc).

Response 2:

This has been added as Supplementary table 2. We have not collected information on ethnicity specifically, although subjects are of white, European decent.

Comment 3:

The authors should assess potential correlation of a composite age-adjusted blood and urine mutation heteroplasmy compared to disease burden/progression (i.e. age-adjusted mean of blood and urine heteroplasmy compared to NMDAS score).

Response 3:

Thank you for this idea. We have carried out this analysis and have amended the table 1 and the text to reflect this.

p.6 “A composite measure of adjusted blood and urine m.3243A>G levels explained no more of the variance than using age-adjusted blood levels alone.”

p.7 “Although the combination of adjusted blood and adjusted urine levels was marginally more associated than adjusted blood levels alone, this difference failed to reach significance (P=0.624).”

Comment 4:

Was m.3243A>G mutation load assessed in buccal swabs or saliva?

Response 4:

No, neither buccal swabs nor saliva are routinely collected from patients who attend the clinic so, although this might be interesting, this analysis was not possible.

Comment 5:

As noted by the authors, only overall disease burden was assessed in this study and a future study of heteroplasmy versus "specific phenotypic features would be valuable". Nevertheless, the authors should consider in the current analysis, muscle heteroplasmy with muscle weakness/functional impairment and urine mutation load with renal involvement.

Response 5:

Thank you for this interesting suggestion, which led us to investigate the association of muscle, urine and blood heteroplasmy with myopathy, ptosis and CPEO within patients who have all three heteroplasmy measures available. We have very few patients with documented renal symptoms or biochemical evidence of renal impairment within the cohort, and so were not able to look at this in detail. We do not see any significant associations with any heteroplasmy measure for ptosis or CPEO. This is likely to be due to a small sample size. We do see a significant association for myopathy with adjusted urine heteroplasmy (P=0.035), but the other heteroplasmy measures are not significant (P_{blood}=0.247, P_{muscle}=0.086). From this, we conclude that larger sample sizes are needed in order to compare the utility of different heteroplasmy measures for individual phenotypes and so we have not included this analysis in the manuscript.

We have looked at the relative roles of heteroplasmy and age in the development of individual m.3243A>G-related phenotypes. This has now been published in *Annals of Clinical and Translational Neurology* (DOI: 10.1002/acn3.532); we have provided a proof of this manuscript as supplementary information.

We have amended the text of the discussion and added this reference to address this comment.

p. 10 “Compensatory mechanisms could well be tissue-specific (we know that heteroplasmy plays a greater role for some tissue-specific phenotypes than for others) and determined by environmental or genetic factors, possibly accounting for some of the phenotypic variability seen in this disease (Pickett et al, 2018).”

Comment 6

In Figure 3B, it appears that the age-adjusted decline in blood heteroplasmy load is due to the effect in women and not in men.

Response 6:

Thank you for noticing; the text now reflects this.

p.5 “There is a relationship between age-adjusted blood heteroplasmy and age (Figure 3B), but this is weaker and only seen in females.”

Comment 7:

The threshold $P < 0.05$ for analysis of variant tests is not appropriate due to the multiple tests performed. A correction for multiple test is warranted.

Response 7:

We agree that we have performed multiple tests, but they are not all independent (due to the correlation between different heteroplasmy measures). We believe that Bonferroni correction is too stringent and results in P values that are not easily interpreted. Therefore, we decided to present unadjusted P values. We have amended the text to make this clear:

p.14-15 “We report unadjusted P values for reasons well-documented in the literature and, particularly as we are testing a priori hypotheses with variables that are not all independent, this would be too conservative (Perneger, 1998).”

Minor Comments:

- 1) Abstract: Change "Age-adjusted blood is the most highly correlated heteroplasmy measure" to "Age-adjusted blood heteroplasmy is the most highly correlated mutation measure"
- 2) Abstract: Change "...indicating that unidentified factors" to "indicating that at least one unidentified factor"
- 3) Results, page 4: Change "exponential" to "continuous".
- 4) Figure citations appear to be incorrect on page 5: "Figure 3C" should be "3B" while "Figure 3B" should be "3C".

Response to minor comments:

Thank you for spotting these. We have made all of these suggested changes.

Responses to Reviewer 2

Summary:

Reviewer 2 is perhaps less reserved in terms of the immediate clinical applications of your work and suggests that at the very least, assessment of mtDNA copy number as a surrogate marker of mt biogenesis, and re-assessment of the statistical models by adding this variable are needed to confer some much-needed molecular and conceptual content to the manuscript.

Comment 1:

On the molecular ground this study lacks one variable that could have been assessed by these experienced group of researchers in mitochondrial medicine, i.e. the mtDNA copy number. Given the feasibility of this experiment, if performed and correlated to the currently existing data, it would increase substantially the novelty.

On the molecular ground, and being the journal of interest EMBO Molecular Medicine, the authors should/could have assessed the mtDNA copy number as surrogate marker of mitochondrial biogenesis, to introduce a further element that may help to interpret the still poor and incomplete correlation of heteroplasmy with disease burden and progression rate. There is clear evidence that heteroplasmy and mtDNA copy number are tightly associated in determine the pathogenicity of the MELAS mutation. Giuseppe Attardi did show this in cybrids a long time ago (Bentlage HA, Attardi

G. Hum Mol Genet. 1996 Feb;5(2):197-205), and recent studies on LHON strikingly highlighted how mtDNA copy number and compensatory activation of mitochondrial biogenesis may determine penetrance (Giordano et al., Brain. 2014 Feb;137(Pt 2):335-53). In the case of MELAS mutation, mtDNA copy number may well modulate clinical expression determining disease burden and rate of progression. Furthermore, the efficiency of compensatory activation of mitochondrial biogenesis that is reflected in mtDNA copy number is individually very variable, with individuals more efficient or less efficient in executing this response. Thus, mtDNA copy number is a molecular variable that seems crucial to fully understand the heteroplasmy issue.

I insist on this point because the authors have the DNA samples and the technical skills to run the experiment. There are issues on having good quality data for mtDNA copy number quantitative assessment based on the DNA extraction methods used, way of DNA conservation etc., all factors well-known to obtain reliable results.

Response 1:

We are very grateful to the reviewer for suggesting that we look at mtDNA copy number; these were experiments that we had planned but agree that the inclusion of this marker of mitochondrial biogenesis has strengthened the models and “*increase[d] substantially the novelty*” of the study.

We have assessed and characterised mtDNA copy number in all three tissues where DNA samples were available and of high enough quality ($N_{\text{muscle}}=66$, $N_{\text{blood}}=197$, $N_{\text{urine}}=165$). Whilst we found no association between disease burden and mtDNA copy number in blood or urine, we did see a highly significant association with muscle mtDNA copy number, which increases the proportion of variability explained to 40%. We have added an additional figure (figure 5) and have amended the abstract, methods (p.14), results (p.7-8) and discussion (p.9-12) to reflect this really interesting finding.

Comment 2:

In conclusion, apart the issue of mtDNA copy number, I believe the authors squeezed as much as possible from the kind of material and data they had available, with appropriate methods and statistical analysis. May be I would have liked to see a bit more reasoning on some of the results. For example, is there any hypothesis to explain the gender difference on urine heteroplasmy, considering that gender recently emerged as a relevant factor in a large survey of Italian patients carrying the same mutation (Mancuso et al., J Neurol. 2014 Mar;261(3):504-10), and is well known in LHON again, possibly correlated with the metabolic effect of estrogens on mitochondrial biogenesis (Giordano et al., Brain. 2011 Jan;134(Pt 1):220-34).

Response 2:

Thank you for recognising the quality of our work. We have expanded our discussion of the gender difference in urine heteroplasmy to reflect this concern.

p.11 “The observed variability and sexual divergence of m.3243A>G heteroplasmy and mtDNA copy number in urinary sediment is likely to be due to differences in cellular content. Urine contains many different cell types, mostly epithelial in nature, and differences in cellular composition arise depending on sex, age and disease status (Benda et al, 2013). Other factors, such as time of day, volume, concentration or infection, could also alter cellular content and therefore influence measured heteroplasmy and copy number levels. If urine is to be pursued for diagnostic testing of m.3243A>G, variability in cellular content and the effect on heteroplasmy levels should be investigated. It is possible that this sexual divergence also exists in other tissues, and may explain some of the sex-specific effects seen in mitochondrial disease (Giordano et al, 2014; Mancuso et al, 2014).”

2nd Editorial Decision

27 March 2018

Thank you for the submission of your revised manuscript to EMBO Molecular Medicine. We have now received the enclosed reports from the referees that were asked to re-assess it. As you will see the reviewers are now globally supportive and I am pleased to inform you that we will be able to accept your manuscript pending the following final amendments:

- 1) Please address referee 1's comments in writing and in the final text. Please provide a point-by-

point letter detailing your answers to the referee and to me.

Please submit your revised manuscript within two weeks. I look forward to seeing a revised form of your manuscript as soon as possible.

***** Reviewer's comments *****

Referee #1 (Comments on Novelty/Model System for Author):

The authors have performed admirable detailed statistical analyses of m.3243A>G mutation load and mtDNA copy number relative to disease severity. The novel finding, that mtDNA copy number has a slight correlation with phenotype, is scientifically interesting, but has modest medical impact as a similar effect has been seen with mtDNA mutations that cause Leber hereditary optic neuropathy.

Referee #1 (Remarks for Author):

The revised manuscript by Grady et al. has been strengthened by the addition of data on mtDNA copy number. As noted by the authors, this information has modestly strengthened the predictive value of their models and increased the novelty of the study. The last sentence of the Abstract is rather tepid and should be enhanced to emphasize the importance of this work. For example, "While our data indicate that age-corrected blood m.3243A>G heteroplasmy is the most convenient and reliable measure for routine clinical assessment, additional factors such as mtDNA copy number appear to influence disease severity."

Referee #2 (Comments on Novelty/Model System for Author):

This study, with the suggested integration of mtDNA copy number assessment, now truly provides a new perspective on the molecular genotype-phenotype correlation.

Referee #2 (Remarks for Author):

I congratulate the authors for having assembled the most advanced molecular analyses now available on the MELAS mutation 3243, having a clearer view of the genotype-phenotype correlation, as obtained by integrating the suggested assessment of mtDNA copy number with mutant load heteroplasmy in the 3 standard tissues, blood-urinary epithelium-skeletal muscle.

2nd Revision - authors' response

28 March 2018

We would like to thank both of the reviewers for taking the time to review our work and for their insightful comments, which have helped improve the quality of the manuscript.

Comments from Referee #1

Comments on Novelty/Model System for Author:

The authors have performed admirable detailed statistical analyses of m.3243A>G mutation load and mtDNA copy number relative to disease severity. The novel finding, that mtDNA copy number has a slight correlation with phenotype, is scientifically interesting, but has modest medical impact as a similar effect has been seen with mtDNA mutations that cause Leber hereditary optic neuropathy.

Remarks for Author:

The revised manuscript by Grady et al. has been strengthened by the addition of data on mtDNA copy number. As noted by the authors, this information has modestly strengthened the predictive value of their models and increased the novelty of the study. The last sentence of the Abstract is rather tepid and should be enhanced to emphasize the importance of this work. For example, "While our data indicate that age-corrected blood m.3243A>G heteroplasmy is the most convenient and

reliable measure for routine clinical assessment, additional factors such as mtDNA copy number appear to influence disease severity."

Response to Referee #1:

Thank you for drawing this to our attention. We have amended this sentence in the abstract to:

“While our data indicate that age-corrected blood m.3243A>G heteroplasmy is the most convenient and reliable measure for routine clinical assessment, additional factors such as mtDNA copy number may also influence disease severity.”

Comments from Referee #2

Comments on Novelty/Model System for Author:

This study, with the suggested integration of mtDNA copy number assessment, now truly provides a new perspective on the molecular genotype-phenotype correlation.

Remarks for Author:

I congratulate the authors for having assembled the most advanced molecular analyses now available on the MELAS mutation 3243, having a clearer view of the genotype-phenotype correlation, as obtained by integrating the suggested assessment of mtDNA copy number with mutant load heteroplasmy in the 3 standard tissues, blood-urinary epithelium-skeletal muscle.

Response to Referee #2:

Thank you for recognising the quality of our work.

Corresponding Author Name: Professor Robert McFarland and Dr. Sarah J Pickett

Manuscript Number: EMM-2017-08262